# Mechanical, Barrier and Thermal Properties of Amylose-Argan Proteins-Based Bioplastics in the Presence of Transglutaminase

**DOI:** 10.3390/ijms24043405

**Published:** 2023-02-08

**Authors:** Michela Famiglietti, Domenico Zannini, Rosa Turco, Loredana Mariniello

**Affiliations:** 1Department of Chemical Sciences, Monte Sant’Angelo Campus, University of Naples “Federico II”, Via Cinthia 4, 80126 Naples, Italy; 2National Research Council, Institute of Polymers, Composites and Biomaterials, Via Campi Flegrei 34, 80078 Pozzuoli, Italy; 3Center for Studies on Bioinspired Agro-Environmental Technology (BAT), University of Naples “Federico II”, Via Cinthia 4, 80126 Naples, Italy

**Keywords:** amylose, argan seed proteins, transglutaminase, mechanical properties, biomaterials

## Abstract

The bioeconomy aims to discover new sources for producing energy and materials and to valorize byproducts that otherwise would get wasted. In this work, we investigate the possibility of producing novel bioplastics, made up of argan seed proteins (APs), extracted from argan oilcake, and amylose (AM), obtained from barley plants through an RNA interference technique. Argan, *Argania spinosa*, is a plant widespread in arid regions of Northern Africa, where it plays a fundamental socio-ecological role. Argan seeds are used to obtain a biologically active and edible oil, producing a byproduct, the oilcake, that is rich in proteins, fibers, and fats, and is generally used as animal food. Recently, argan oilcakes have been attracting attention as a waste to be recovered to obtain high-added-value products. Here, APs were chosen to test the performance of blended bioplastics with AM, because they have the potential to improve the properties of the final product. High-AM-starches present attractive features for use as bioplastics, including a higher gel-forming capacity, a higher thermal stability, and reduced swelling compared to normal starch. It has already been demonstrated that pure AM-based films provide more suitable properties than normal starch-based films. Here, we report on the performance of these novel blended bioplastics in terms of their mechanical, barrier, and thermal properties; and the effect of the enzyme microbial transglutaminase (mTGase) as a reticulating agent for AP’s components was also studied. These results contribute to the development of novel sustainable bioplastics with improved properties and confirm the possibility of valorizing the byproduct, APs, using them as a new raw material.

## 1. Introduction

One of today’s most crucial environmental challenges is solving plastic pollution. With a compound annual growth rate (CAGR) of 8.4%, since the 1950s plastic production has been increasing faster than any other class of manufactured materials [1]. It is estimated that plastic synthesis is going to require half of the global oil demand by 2050, with emissions of over 56 billion Mt of carbon dioxide equivalent (CO_2e_), around 10–13% of the remaining carbon budget [2]. Plastics are derived from fossil fuels, and each stage of their life, from extraction up to their end-of-life, produces emissions of greenhouse gases (GHGs). Moreover, their molecular structures exhibit strong resistance to environmental degradation, producing materials that persist for decades or longer if not destroyed by incineration [3]. Some of the strategies to reduce the global carbon footprint of plastics are replacing fossil fuel-based plastics with biobased plastics, increasing their recycling rate, and reducing their growth in demand [4]. Biobased plastics are derived from renewable plant feedstocks, and in their overall life cycle produce lower GHG emissions than conventional plastics [5]. However, this is dependent on their composition, their end-of-life management, and the carbon storage potential lost because of the lands needed to cultivate the biomass, with damage to biodiversity and a loss of soil dedicated to food production [4]. Other challenges to be overcome are their poor performances, variability of feedstock properties, and high production costs [6].

Starch is one of the most attractive biopolymers to produce bioplastics because of its abundance (it is the predominant carbohydrate reserve in plants), cheapness, and its well-defined chemical features [7]. Starch consists of two glucose-based polymers, amylose (AM) and amylopectin. In the native starch granule, AM is mainly amorphous and comprises predominantly linear chains of α-1,4-linked glucose residues, with a molecular weight (M_w_) of 10^5^ g/mol, whereas amylopectin exists as a semicrystalline highly branched α-1,4: α-1,6 glucan polymer with a M_w_ of 10^6^–10^7^ g/mol [8]. Normally, starch is made up of about 20–30% AM and 70–80% amylopectin, but the AM content varies from less than 5% in waxy starches up to 70% in high-AM starches [9,10]. The amylopectin molecule contains regions with low and high levels of branching. In the highly branched regions, side-chain branches intertwine to form double helices, whose association gives rise to the crystalline zones. Three types of crystalline structure have been identified as A-type, B-type, and C-type, with the latter consisting of a mixture of A and B types. In native starch, AM is mainly amorphous, but if starch granules are subjected to gelatinization in aqueous media, the AM adopts a single left-handed helical conformation, called Vh-type, with a pitch of 8 Å, where ligands like hydrophobic molecules or hydrophobic side chains of molecules can arrange. AM-lipid complexes can be endogenously present in starch or formed after gelatinization in the presence of added lipids, as during starch-processing techniques [11].

The structure of starch granules present an amorphous bulk core area surrounded by concentric semicrystalline growth rings alternating with amorphous growth rings. Starch granules can be examined on at least five levels of structure. At the largest scale are the intact granules, with sizes from 1 to 100 μm. Next come the alternating semicrystalline and amorphous growth rings. Then come the block structural elements that are made up of the left-handed superhelices, and finally there are the structural elements of superhelices that are crystalline and amorphous lamellae [12]. During high-temperature processing, starch granules undergo structural changes that determine some functional properties such as water uptake, swelling of the granule, formation of a viscoelastic paste, reassociation of starch chains, and formation of a gel. When starch is heated in water, the granules become hydrated and swollen, transforming into a paste. This is caused by the breaking of hydrogen bonds and the unwinding of the double helix, resulting in the final collapse of the granule. During the cooling, starch chains go to meet a phenomenon called retrogradation, in which they return to a partially ordered structure that, nevertheless, differs from the original structure. This induces some further physical changes in viscosity, gel formation, and degree of crystallinity. Starch retrogradation occurs because of intermolecular hydrogen-bond formation between residues of AM molecules and residues of sidechains of amylopectin molecules, and between different molecules of amylopectin. This process of reassociation and recrystallization can vary for different types of starch because it depends on the AM and amylopectin contents. For bioplastic purposes, these phenomena have a crucial relevance: the formation of crystallites affects the mechanical properties, causing a decrease of elongation at break and an increase in tensile strength. The films become stiffer, less flexible, and more difficult to handle [12]. Even if AM retrogradation was found to be a fast event relative to amylopectin retrogradation, it is worth noting that AM chains longer than approximately 1000 glucose units form a gel consisting of crosslinked AM chains [13]. Thus, high AM starches present interesting features for bioplastics production, including a high gel-forming capacity, enhanced mechanical properties, a high thermal stability, and reduced swelling. These improvements are attributed to the entanglement of long linear AM chains and the partial retention of granular structures, which act as self-reinforcement [14].

Thus, modifying the ratio of AM/amylopectin leads to changes in functionalities of the starch. Until now the separation of AM from amylopectin has not been performed on a significant production scale because of the high process cost. Innovative systems to provide added-value functionalities directly in the crop, also known as molecular farming, can be used to improve the yields and reduce the costs of production [7]. Classical breeding, random mutagenesis, and site-directed mutagenesis technology have already provided many important special starch qualities [15]. The present work utilizes AM derived from a transgenic barley line that synthesizes starch with 99% of AM. This line was obtained by the RNA interference technique, silencing genes coding for starch branching enzymes (SBE I, SBE IIa, and SBE IIb) [16]. AM derived from this crop line has already been used to produce biomaterials: Sagnelli et al. tested a crosslinked AM in the presence of glycerol, and Xu et al. verified the performance of an AM blended with cellulose nanofibers using glycerol as a plasticizer [13,15].

The aim of this work is to contribute to overcoming the challenges to new biobased plastics in terms of the performance of materials, the cost of feedstocks, and the sustainability of the entire process of production. The choice to develop a blended biomaterial derives from the effort to reduce the cost of production, avoid the loss of carbon storage potential, utilize an inexpensive byproduct as a resource that otherwise would be destined for disposal, and improve the bioplastic properties. Recent research has focused on the recovery of waste material from oil seed processing of plants such as hemp, cardoon, cottonseed, sunflower, and argan [17,18,19]. These crops are grown all over the world, mainly for oil extraction, which produces a waste around 50% of the original seed weight. This byproduct, called oil seed cake, is made up of a high percentage of proteins besides fibers, remaining fats, and different bioactive compounds, such as antioxidants, vitamins, and minerals [20].

Argan, *Argania spinosa,* is an endemic species of Morocco and the southwest Algeria region belonging to the *Sapotaceae* family [20]. It can be cultivated in arid and semi-arid regions and protects soils from desertification, erosion, and sand encroachment thanks to its long and deep roots; its wood is used as fuel by the local population and its leaves are usually exploited as forage [21]. Argan fruits are a source of biologically active and edible oil that finds applications in cosmetic and pharmaceutical products or is consumed as food. It is rich in monounsaturated and polyunsaturated fatty acids, and in minor compounds such as tocopherols, polyphenols, and carotenoids [22]. The byproduct derived from the argan oil extraction process, referred to as argan press cake (APC), has traditionally been used for animal feeding because of its high levels of dry matter, that includes proteins (48.4%). APC is also rich in fibers (17.6%) and fats (18.9%), and contains significant levels of calcium, potassium, and phosphorous [23]. Recently, Mirpoor et al. used argan seed proteins (APs) derived from APC to produce protein-based bioplastics by the casting method and using glycerol as a plasticizer. They showed satisfying mechanical, barrier, and hydrophilicity features, and they were successfully tested as wound dressings [19]. Here, APs were chosen to test the performance of a blended bioplastic with AM, because they represent a waste to be recovered, that does not subtract soil from food cultivations, and that has the potential to improve the properties of the final product. Moreover, it was demonstrated that APs act as a substrate in microbial transglutaminase (mTGase, E.C.2.3.2.13)-mediated polymerization. In recent years, the use of enzymes as a biotechnological tool for strengthening the film polymer matrix has increased [24]. Using chemical crosslinking reagents is becoming less acceptable because of their potential harmfulness and toxicity, especially in the production of edible films. The use of mTGase has been proposed for its capability to catalyze intra and/or intermolecular isopeptide bonds between the γ-carboxamide group of glutamine (acyl donor) and the ε-amino group of lysine residues (acyl acceptor). Proteins modified with mTGase showed changes in their functional characteristics, influencing the properties of film matrices [25].

## 2. Results and Discussion

### 2.1. Transglutaminase Assay

The use of mTGase has been proposed for its capability to act as a biotechnological tool for strengthening the film polymer matrix, catalyzing intra and/or intermolecular isopeptide bonds between the γ-carboxamide group of glutamine residues (acyl-donor) and the ε-amino group of lysine residues (acyl-acceptor).

To verify the effect of mTGase on APs in the presence of AM, the SDS-PAGE profile following protein incubation with mTGase (40 U/g) was carried out. Figure 1 shows that APs act as both acyl donor and acyl acceptor substrates of the enzyme. APs samples incubated in the absence of mTGase exhibited three main bands and two less intense bands (Figure 1 (1,2)). A decrease in the intensity of the low molecular protein bands and the formation of high molecular weight polymers (Figure 1 (3,4)), that are not able to enter the stacking gel and appear at the entrance of the wells, demonstrate the effect of the enzyme.

### 2.2. Two-Dimensional Polyacrylamide Gel Electrophoresis (2-D PAGE)

The APs analysis was carried out using 2-D PAGE in the absence and presence of mTGase (40 U/g). As shown in Figure 2 panel A, four main groups were observed in the absence of the enzyme with pI nearby neutral or alkaline values and Mw in the range 15–75 kDa. The effect of the enzyme is evident in panel B, since the intensity of spots has decreased for all the groups identified, confirming the role of APs as a substrate for mTGase. APs subjected to the crosslinking reaction present a higher Mw, that does not allow them to penetrate the gel: a less severe spot is visible to the interface of the wells.

### 2.3. Zeta Potential Analysis of Film Forming Solutions (FFSs) Prepared Blending AM and APs in the Absence and the Presence of mTGase

Different FFSs were prepared, with different ratios APs-AM (0/100–15/85–30/70–50/50–100/0% (*w/w*)), and their stability was investigated using the Zeta Potential analysis (Table 1). The first sample (containing 100% AM) exhibits a Zeta Potential value of −10.85 ± 1.39 mV, indicating a low stability of FFSs. This result can be explained by taking into account that, during the experiment, the sample goes through a recrystallization process due to FFSs cooling. At 50 °C (our casting T) water evaporates quickly, promoting hydrogen bond formation among AM chains, which results in a well-structured matrix. Increased Zeta Potential values can be observed in all the other samples, increasing the protein content because the presence of proteins influences the interactions among AM chains, preferring interactions between proteins and AM. Lower values of Zeta Potential in the presence of the enzyme could be due to increased interparticle interactions (Van der Waals, hydrophobic interactions together with hydrogen bonding) among proteins modified using the enzyme.

### 2.4. Opacity, Density, Thickness, and Morphology

Values of opacity, density, and thickness of the films with an increasing amount of Aps, incubated in the absence and the presence of mTGase, are shown in Table 2. The optical properties are crucial to developing products, such as for food packaging, able to attract the interest of consumers. As reported, AM100-based films possess an opacity value of 1.84 ± 0.34 A_600nm_/mm, similar to the commercial petrol-based plastic LDPE (low-density polyethylene) (opacity value = 1.44 ± 0.04 A_600nm_/mm) [19]. The addition of APs notably increases the opacity of the films already, with a content of 15% (*w/w*). The film color becomes darker proportionally with the increase of APs content (Figure 3). It reaches the highest value, for APs100-based films, of 5.55 ± 0.18 A_600nm_/mm that, nevertheless, is lower than the opacity value of commercial MaterBi^®^ (61.92 ± 3.55 A_600nm_/mm) [19]. The presence of mTGase reduces film opacity: probably, crosslinking reactions induce the formation of a more ordered structure of film matrices, increasing light transmission and changing the refractive index. Indeed, the opacity value was significantly lower in APs100-based films treated with mTGase (4.79 ± 0.04) than in APs100-based films cast without the enzyme, even if the latter were significantly thicker than the first. Thickness increases with the increasing of APs content, and further in the presence of mTGase when the amount of APs reaches 50% (*w/w*). Density values are not significantly different among the samples treated or not with the enzyme.

Film microstructure was determined by SEM-analysis: film surface and cross-section images are reported in Figure 4. The microstructure of the films depends on the interactions among the films’ components that directly affect their physical, mechanical, and barrier properties. SEM surface analysis of AM100-based films, blended films (APs50-AM50), and APs100-based films, incubated in the absence and the presence of mTGase, showed that the topography changed significantly. The AM100-based film surface was smooth (Figure 4A), compared to the surface of a pure AM-based film prepared by Xu et al., in the absence of glycerol, which showed pleated structures [13]. The plasticizer used to prepare the AM-based films, creating new hydrogen bonds, filled the free volume among the polysaccharide molecules making the surface homogenous. The blended films showed a less smooth and less homogeneous surface, probably due to the presence of the oil fraction in APs, as reported also by Mirpoor et al. [19], both in the presence and absence of mTGase (Figure 4B,C). The cross-sections of blended films (Figure 4B,C) showed irregularities and cracks that visibly increased in APs100-based films (Figure 4D,E). Formation of a phase separate from the aqueous phase resulted in oil droplets after drying that made the surface heterogeneous and discontinuous, both for films treated or not with the enzyme (Figure 4D,E).

### 2.5. Film Mechanical Properties

As already reported, AM can improve film performance in terms of mechanical properties, relative to normal starch, because it influences the degree of crystallinity and the entanglement of AM chains [13,15]. This results in an increase in the Tensile Strength (TS) and a decrease in the Elongation at Break (EB). This latter direct effect can be overcome using a plasticizer such as glycerol [14].

In this work, the same content of glycerol was used for all the produced films. AM100-based films show the highest EB (~70%), with respect to both blended films and APs100-based films, a value of TS similar to APs100-based films (~4.6 and ~5 MPa respectively) and higher than blended films, and a value of Young’s Modulus (YM) lower than APs100-based films (~100 MPa and ~310 MPa respectively) and similar to blended films. As reported in Figure 5, AM100-based films show preferable mechanical characteristics since they are extensible, not stiff, and at the same time resistant. Blended films exhibit a TS value lower than both AM100-based films and APs100-based films, an EB value lower than AM100-based films but higher than APs100-based films, and a YM value lower than APs100-based films and similar to that of AM100-based films. The AM component seems to keep the extensibility and the lower stiffness in blended films, but the entanglement of AM chains is probably interrupted by the presence of Aps, resulting in reduced values of TS, EB, and YM with respect to the AM100-based films. The presence of mTGase reduces the TS and the YM of APs100-based films, increasing their EB, this effect has already been noted for other protein-based films modified by mTGase [26].

### 2.6. Moisture Content, Water Solubility, Swelling Ratio

Figure 6 shows further results for the moisture content, water solubility, and swelling ratio of AM100-based films, APs100-based films in the absence and presence of mTGase, and blended films (APs50-AM50) treated or not with the enzyme. The moisture content of the AM100-based films was significantly higher than the blended films and the APs100-based films, as confirmed also by FT-IR analysis, by which the presence of free water molecules is detected. Moreover, the moisture content slightly decreases in the presence of mTGase: probably this effect is due to the reduction of the free ɛ-amino groups of lysine residues after the formation of the isopeptide bond catalyzed by mTGase [17]. Conversely, the film water solubility increases with the addition of APs: blended films and APs100-based films show similar water solubility values, higher than the value of AM100-based films. Similarly, the swelling ratio is higher for blended films and further increases for APs100-based films relative to AM100-based films. The entanglement of AM chains can explain this effect, because the hydroxyl groups are not available to bond with free water molecules resulting in reduced swelling ratio and solubility. In contrast, the hydrophilic nature of APs probably promotes the absorption of free water molecules, promoting the swelling of the films and making them more soluble.

### 2.7. Water Vapor Permeability (WVP) and CO_2_ Permeability (CO_2_P)

Water vapor and CO_2_ permeabilities were investigated because of their crucial importance for film applications such as food packaging (Table 3). Film barrier properties influence the shelf life of food products and can improve food quality during storage.

To evaluate the permeability function of hydrocolloid-based films, some parameters have to be taken into account: thickness, crystallinity, porosity, plasticizer content, and water activity. Usually, plasticized-starch-based films show poor water vapor barrier properties because of their hydrophilic nature. Here, AM100-based films showed a WVP of 8.7 g mm/m^2^ d kPa, similar to that exhibited by high-AM-based films reported by Colussi et al. [27] and by the commercial starch-based bioplastics Mater Bi (9.8 g mm/m^2^ d kPa) [17]. Addition of APs reduced proportionally the WVP, probably because of the decreasing of the mass transfer rate through the film of water vapor molecules that bind the hydrophilic groups of APs, as demonstrated using the investigation of the swelling ratio. Nevertheless, the WVP value for APs100-based films is still far from that exhibited by commercial conventional plastics such as LDPE (0.075 g mm/m^2^ d kPa). The effect of mTGase confirmed the results already reported in different papers regarding the improved film barrier capability, probably due to the decrease of the available free polar groups of APs following the formation of isopeptide bonds catalyzed by the enzyme [17].

AM100-based films show stronger CO_2_ barrier capability, equal to 0.42 ± 0.04 cm^3^*mm*m^−2^*d^−1^*kPa^−1^, than the commercialized bioplastics MaterBi^®^ and LDPE (5.19 ± 0.60 and 13.99 ± 1.08 cm^3^*mm*m^−2^*d^−1^*kPa^−1^, respectively). Furthermore, blended films exhibit improved barrier capability relative to both AM100-based films and APs100-based films. In the presence of mTGase, this effect seems to be conserved: blended films show a lower value of CO_2_ permeability than pure films of AM and APs treated by the enzyme.

### 2.8. Fourier Transform Infrared Spectra (FT-IR)

The AM100-based film’s spectrum (Figure 7) shows a wide absorption band at 3000–3600 cm^−1^, attributed to the stretching vibration of O-H bonds, which are the dominant functional group and are involved in intra- and inter-molecular hydrogen bonds with other hydroxyl groups [28]. The peak at 3272 cm^−1^ is due to the strong hydrogen bond formation between the hydroxyl groups in the AM chains and the hydroxyl groups in the glycerol molecules. This new and strong hydrogen bond formation between AM and glycerol replaces the original interactions in the AM chains, shifting to lower wavenumbers compared to the peak detected for AM in the absence of glycerol, as reported in Muscat D. et al. [28]. The AM100-based film’s spectrum exhibited a peak at 1647 cm^−1^ corresponding to the presence of free water molecules (water molecule bending) [29]. The peak at 1337 cm^−1^ in the AM100-based film’s spectrum is associated with the O-H bending of primary or secondary alcohols [28]. Finally, the area between 1200 and 950 cm^−1^ is the fingerprinting region: peaks at 920, 996, and 1075 cm^−1^ are associated with the C-O stretching in the anhydro-glucose ring, while a peak at 1149 cm^−1^ is assigned to C-O-C asymmetric bending [30,31]. The characteristic C-C and C-O vibration bands of glycerol, normally occurring in the range 850–1100 cm^−1^, are probably overlapped by signals of AM and APs, and not easily detected, as reported also by Simona J. et al. [30].

The broad band 3600–3000 cm^−1^ peak for the APs100-based film in the presence or absence of mTGase (Figure 8) was attributed to stretching vibrations of –OH and –NH groups, but its intensity decreases as compared to the AM100-based film’s spectrum because of the lower presence of -OH groups. Asymmetric and symmetric stretches of CH_2_ groups were identified around 2920 and 2850 cm^−1^. These peaks are related to the fats present in APs: in fact, because of the oil remaining in the argan oilcake, this area of the spectrum is higher in absorbance in APs100-based films. The sharp peak at 2924 cm^−1^ was observed also in AM100-based films, followed by a shoulder peak at around 2850 cm^−1^ due to the presence of lipids/AM complexes [15].

The peak at 1745 cm^−1^ is attributed to the stretching vibration of ester carbonyl C=O functional groups of the triglycerides present in APs. The area of the spectrum between 1480–1200 cm^−1^ is the fingerprint area of proteins, which was attributed to single bond (C-H and N-H) vibration and tautomerism of the amide structure [32].

The absorption bands at 1630 and 1540 cm^−1^ represent amides I and II, respectively. The presence of a peptide bond is usually indicated by a C-O vibration at the amide I band (1700–1600 cm^−1^) and N-H bending vibration and C-N stretching vibration at the amide II band (1600–1500 cm^−1^). Generally, the amide I band indicates the secondary structure of the protein, in particular, absorption at 1610–1640 cm^−1^ can be assigned to the β-sheet, while the amide II band represents the environment for hydrogen bonding [33].

Despite the presence of AM, in the blended film spectra there was no significant shift in the absorption peaks of the amides I and II (Figure 9). This shows that the addition of AM had no significant influence on the secondary structure of the proteins, as reported also by Xu X. et al. [31]. The blended film’s spectrum shows the characteristic peaks of AM and APs without significant shifts, indicating that interactions, if any, cannot be detected this way.

In the same way, mTGase does not seem to cause changes in the structure of the proteins detectable by FT-IR analysis.

### 2.9. Thermogravimetric Analysis (TGA)

The TGA and the differential thermogravimetric (DTG) curves obtained are reported in Figure 10. All the curves were normalized with respect to the starting sample size. From the AM100-based film’s thermogram three different regions of weight loss can be detected (Figure 10a). In the first one, from 30 °C up to 100 °C, the sample lost about 10–15% of its weight, which was attributed to the free and bound water molecules in the films. The second mass loss process was observed in the range of 100–200 °C, which is the characteristic event of polysaccharides and is also associated with the thermal degradation of glycerol [34]. The third stage, between 300 and 350 °C, was possibly associated with the main degradation of starch crystallites containing AM/lipid complexes [35]. The DTG thermogram (Figure 10b) for APs100-based films in the absence of mTGase showed, besides the mass loss associated with the release of moisture, three main regions of weight loss. The first one, between 180 and 200 °C, was probably due to the depolymerization of low molecular weight proteins [31]. The second one, from 200 °C to 260 °C, could be due to the decomposition of hemicellulose, possibly present in APs, and the third peak, which appears as a pronounced shoulder, includes the thermal degradation of cellulose and lignin present in APs in the range between 320 and 350 °C [36]. APs100-based films treated with mTGase showed some differences: it seems that the enzyme improves the thermal stability of the protein component of APs-based films, shifting their maximum degradation temperature to ~240 °C. This value may be attributed to the crosslinked polymerization reaction catalyzed by the enzyme that results in a stiffer network with a higher thermal stability. Still, in this film sample, only a second region of weight loss can be detected (between 270 and 350 °C), which probably represents the degradation of the lignocellulose fraction in the presence of the mTGase-modified proteins [37]. Differently, blended films (AM50-APs50) treated with mTGase showed a clear decrease in the maximum degradation temperature at around 170 °C, a value lower than the one registered in the sample prepared in the absence of the enzyme (200 °C), as reported in Figure 10b. This could be due to the polymerization reaction (isopeptide bonds) mediated by mTGase, that results in the formation of high molecular weight proteins that interrupt the entanglement of AM chains and create discontinuous domains, making films more susceptible to thermal degradation.

For the same reason probably, in both blended films, treated or not by mTGase, the peaks related to AM were observed at lower temperatures (280–285 °C) than the AM pure single peak of AM100-based films. Moreover, the reduction of the thermal stabilities could be caused by the presence of APs oil that, as a plasticizer, increases the macromolecular mobility, encouraging AM chains’ thermal degradation, as widely reported in the literature [38].

Finally, even in the blended films, it is possible to observe a clear and small peak in the range of 300–400 °C, caused by the thermal degradation of the cellulose fraction.

### 2.10. Differential Scanning Calorimetry (DSC)

In the DSC thermograms reported in Figure 11, only the melting temperature can be observed. There is no glass transition temperature, indicating that amylose, like starch, undergoes degradation. AM100-based films show an endothermic peak around 127 °C, which is the melting temperature of AM/lipid complexes and is slightly higher than the melting temperature of LDPE (105–123 °C) [39]. The endothermic profile of AM transition was in accordance with the one reported by Sagnelli et al. [40]. DSC results indicated a higher melting temperature for APs100 (endothermic peak at 150 °C), related to the denaturation of APs. For blended films treated with the enzyme, a second transition was found in the temperature range of 90–105 °C. This probably corresponds to the denaturation of low molecular weight proteins because, as reported in the literature, mTGase-mediated polymerization destabilizes non-covalent interactions within the proteins [41].

## 3. Materials and Methods

### 3.1. Materials

Argan seeds were purchased from a local market in Marrakech (Morocco). AM was produced by Professor Andreas Blennow’s team at the University of Copenhagen. Chemical reagents used for electrophoresis were purchased from Bio-Rad (Segrate, Milano, Italy).

mTGase (Activa) was supplied by Prodotti Gianni (Milano, Italy).

Sodium hydroxide, hydrochloric acid, and n-hexane (99%) were purchased from Sigma Aldrich Company (St. Louis, MO, USA). Glycerol and magnesium nitrate were purchased from Carlo Erba S.p.A. (Milan, Italy).

### 3.2. Preparation of APs Concentrate

APs were extracted according to the method described by Mirpoor et al. [19]: argan seeds were ground in a miller (Retsh GH200) for 3 min and the oil was extracted by means of the Soxhlet apparatus using n-hexane as solvent. The sample was then kept in an oven at 45 °C to let the solvent evaporate. After that 100 g of the sample was dissolved in 1 L of distilled H_2_O and the pH was adjusted to 11 by NaOH 1 M, then stirring the mixture for 1 h. The supernatant was collected after centrifugation at 10,000× *g* for 30 min and the pH was adjusted to 5.4, to allow the precipitation of proteins. Centrifugation at 10,000× *g* for 30 min followed; the pellet was collected and dried at 25 °C and 45% RH for 2 days to obtain the APs concentrate. Finally, the latter was ground in a mortar and the protein content was determined by Kijeldhal’s method, using a nitrogen conversion factor of 6.25.

### 3.3. Transglutaminase Assay

SDS-PAGE was used to verify the action of mTGase on APs, also in the presence of AM. Film-forming solutions with the same amount of APs and AM (50% (*w/w*) APs and 50% (*w/w*) AM) were prepared, as described below, and incubated at 37 °C in the presence of 40 U/g of proteins. 50 µL of sample buffer (15.5 mM Tris-HCl, pH 6.8, 0.5% (*w/v*) SDS, 2.5% (*v/v*) glycerol, 200 mM β-mercaptoethanol and 0.003% (*w/v*) bromophenol blue) were added to the reaction mixtures at the end of the incubation. Samples containing 30 and 60 µg of proteins were heated for 5 min in a boiling water bath and analysed by SDS-PAGE (Precast SDS-PAGE gel 12% Mini-Protein gels, Bio-Rad, Segrate (MI), Italy)) at 80 mA for 2 h. Bio-Rad Precision Protein Standards were run as molecular weight markers.

### 3.4. Two-Dimensional Polyacrylamide Gel Electrophoresis (2-D PAGE)

2-D PAGE was used for the detection and analysis of APs. In the first dimension (IEF isoelectric-focusing PAGE), proteins were separated by the pI value, and in the second dimension (SDS-PAGE) by the relative molecular weight. IEF was carried out using 7 cm IPG strips (pH 3–10) (Bio-Rad ReadyStrip IPG) and analyzing 100 µg of APs previously dissolved in 125 µL of rehydration sample buffer (Bio-Rad ReadyPrep 2-D Starter Kit Rehydratation/Sample Buffer). The sample was loaded in an IEF cell and 2 mL of mineral oil were added to the strip to avoid evaporation during the 24 h protein separation in a Bio-Rad PROTEAN IEF CELL, with a potential difference of 50 V. Before the SDS-PAGE step, oil was removed and the strip was treated with DTT and iodoacetamide. Precast SDS-PAGE gel (12%, Mini-Protein gels, Bio-Rad, Segrate (MI), Italy) was used to carry out the SDS-PAGE at 80 mA for 40 min. The gel was finally stained with Coomassie Brilliant Blue R250 (Bio-Rad).

### 3.5. Preparation of FFSs

Different FFSs were prepared with different ratios of APs-AM (0/100–15/85–30/70–50/50–100/0 (%*w/w*)) using H_2_O as a solvent to have a final concentration of 10 mg/mL, and glycerol as plasticizer with a concentration of 50% (*w/w*) (in respect to the total mass). AM was gelatinized by leaving it to hydrate overnight at 4 °C and then heating it in an oil bath at 140 °C for 1 h using a hydrothermal autoclave reactor. APs were solubilized in H_2_O adjusting the pH to 12 with NaOH 1M and stirring the solution for 1 h. FFSs were obtained by mixing AM after gelatinization and APs after solubilization in the correct ratios.

### 3.6. Zeta Potential

The Zeta Potential of the FFSs prepared with different ratios of APs-AM (0/100–15/85–30/70–50/50–100/0 (%*w/w*)) were analyzed using the Zetasizer Nano-ZSP (Malvern^®^, Worcestershire, UK).

The Zetasizer Nano combines different techniques of light scattering to obtain a complete characterization of a colloidal system. Operating with a helium-neon laser at a fixed wavelength of 633 nm, it determines the Zeta Potential by using the Electrophoretic Light Scattering (ELS).

Three independent measurements were carried out on each sample, diluted to have a final concentration of 1 mg/mL.

### 3.7. Films Preparation

FFSs were poured into polypropylene Petri dishes (9 cm in diameter) and dried in an oven at 50 °C for ~28 h. After that, the films were stored in a desiccator at room temperature to balance the moisture content at ~50% RH for 4 days before the subsequent analyses. The appearance of the films is shown in Figure 3.

### 3.8. Film Characterization

#### 3.8.1. Film Transparency

The opacity of the films was determined according to Jahed et al. [42]. The method involved the measurement of the absorbance of the films at a wavelength of 600 nm. Each film was cut into 1 cm × 4 cm strips, placed in a quartz cuvette, and forced to adhere to its wall, to measure the absorbance by using a UV-Vis spectrophotometer (SmartSpec 3000 Bio-Rad, Segrate, Milan, Italy). The opacity value was then obtained by calculating:Opacity = A_(600nm)_/X_(mm)_
where A_(600 nm)_ is the absorbance and X_(mm)_ is the film thickness, determined as the average in five spots on each specimen by means of a digital micrometer (IP65 Alpa Exacto, Alpa metrology Co., Pontoglio (BS), Italy) with a precision of 0.001 mm. Three randomly chosen specimens of each sample were examined.

#### 3.8.2. Film Density

To determine the density (g/cm^3^) of the films, each sample was cut into 2 × 2 cm^2^ pieces and weighed after conditioning. The film density value was calculated according to the following equation reported by Cruz-Diaz [43]:ρ= m_(g)_/A_(cm_^2^_)_ X_(cm)_
where m is the dry mass of the film, A is the film area, and X is the film thickness, determined as described above. Three specimens of each sample were randomly examined.

#### 3.8.3. Morphology

The morphology of the films was examined using field emission scanning electron microscopy (SEM) (Nova NanoSem 450-FEI-Thermo Fisher, Scientific, Waltham, MA, USA). The films were cryo-fractured in liquid nitrogen and then the samples were coated with a thin layer of Au-Pd using a vacuum sputter coater. They were finally observed at a magnification of 2000× with an accelerating voltage of 5 kV by using an ETD detector.

#### 3.8.4. Film Mechanical Properties

The Tensile Strength, Young’s Modulus, and Elongation at Break were determined by using a dynamometer (Instron universal testing instrument model no. 5543A, Instron Engineering Corp., Norwood, MA, USA) according to ASTM D882-18 (1997). Each film was cut into strips with a length of 40 mm and a width of 10 mm, and they were tested by using a 1 kN load cell with a rate of grip separation of 10 mm/min. The film thickness was determined in five spots on each specimen by means of a digital micrometer (IP65 Alpa Exacto, Alpa metrology Co., Pontoglio (BS), Italy) with a precision of 0.001 mm.

#### 3.8.5. CO_2_ and Water Vapor Permeability

Film barrier permeabilities to CO_2_ and water vapor were analyzed using a MultiPerm apparatus (ExtraSolution s.r.l, Pisa, Italy) according to the Standard Methods (ASTM D3985-05, 2010; ASTM F-2476-13, 2013). Each sample is loaded within the instrument, where it constitutes a separate septum between two semi-chambers. A stream of the permeant gas is made to flow through the upper chamber and permeate through the sample, then it is picked up by the carrier gas and detected by a sensor. This process is carried out while maintaining the chamber at a fixed temperature (25 °C) and monitoring continuously the relative humidity (50%), the flow, and other variables that can alter the permeation of the sample. Aluminum masks were used to reduce the film test area to 2 cm^2^, and the analyses were performed in duplicate.

#### 3.8.6. Moisture Content and Solubility

Film moisture content and solubility were determined according to the method described by Zahedi et al., with some modifications [44]. The analysis was performed in triplicate on samples of 2 cm^2^.

Each sample was initially weighed (W_i_), dried at 105 °C in an oven for 24 h, and after drying weighed again (W_d_). The moisture content was determined by calculating the difference between the initial and the final weight of the samples using the following equation:Moisture content (%) = [(W_i_ − W_d_)/W_i_] × 100

To determine film solubility, the dried samples were then immersed in 30 mL of distilled H_2_O and stirred at 25 °C for 24 h in a shaker incubator. After that, the undissolved films were carefully collected and dried in an oven at 105 °C for 24 h. The final weight (W_f_) of the dried samples was measured, and the solubility value was calculated as follows:Water solubility (%) = [(W_i_ − W_f_)/W_i_] × 100

#### 3.8.7. Swelling Ratio

The swelling ratio of the films was determined by weighing the film samples (2 cm^2^) (W_i_) before immersing them in 30 mL of distilled H_2_O at 25 °C for 1 h. The films were then collected, dried with absorbent paper, and weighed again (W_s_). The film swelling ratio was calculated using the following equation as reported by Roy S. et al. [45]:Swelling ratio (%) = [(W_s_ − W_i_)/W_i_] × 100

The analysis was carried out in triplicate.

#### 3.8.8. Fourier Transform Infrared Spectra (FT-IR)

FT-IR analysis was performed at room temperature by using an FT-IR Nicolet 5700 spectrophotometer (Thermo Fisher Scientific, Waltham, MA, USA). The FT-IR spectrum of each sample was recorded in the range of 4000–500 cm^−^^1^ (spectral resolution of 2 cm^−^^1^, 64 average scans) and processed using the Omnic software.

#### 3.8.9. Thermogravimetric Analyses (TGA)

Thermogravimetric analyses were carried out with a Perkin-Elmer Pyris Diamond thermogravimetric analyzer (TGA/DTA), equipped with a gas station. An amount of 3–4 mg of each sample was placed in an open ceramic crucible and heated from 30 °C up to 600 °C at a rate of 10 °C/min, under nitrogen at 30 mL/min.

#### 3.8.10. Differential Scanning Calorimetry (DSC)

The melting temperature of the samples was determined by Differential Scanning Calorimetry (DSC) using a Q2000 T zero DSC, TA Instrument (New Castle, DE, USA), equipped with a liquid nitrogen accessory for fast cooling. The calorimeter was calibrated for temperature and energy using indium. Dry nitrogen was used as the purge gas. A single scan was run at 10 °C/min from 25 to 250 °C. The melting temperature was defined from the endotherm peak value.

### 3.9. Statistical Analysis

The SPSS19 (Version 19, SPSS Inc., Chicago, IL, USA) software was used for all statistical analyses. One-way analysis of variance (ANOVA) and Duncan’s multiple range tests (*p* < 0.05) were used to determine the significant difference among the samples.

## 4. Conclusions

Engineered amylose (AM) was used to prepare argan protein-based (APs) composite films. Their characteristics were studied with respect to different protein concentrations and compared to AM-based and APs-based films. The protein content seems to influence Elongation at Break giving rise to less extensible films. In the presence of the enzyme, this property is significantly lowered, making these blended films interesting for application as bio-shoppers. The presence of proteins also influences the water vapor permeability in composite films, providing a higher barrier effect which notably increases when TGase is used as a reticulating agent. Even though thermal analysis results suggest that the thermal properties are not greatly influenced by film composition, all samples analyzed show good thermal stabilities.

## Figures and Tables

**Figure 1 ijms-24-03405-f001:**
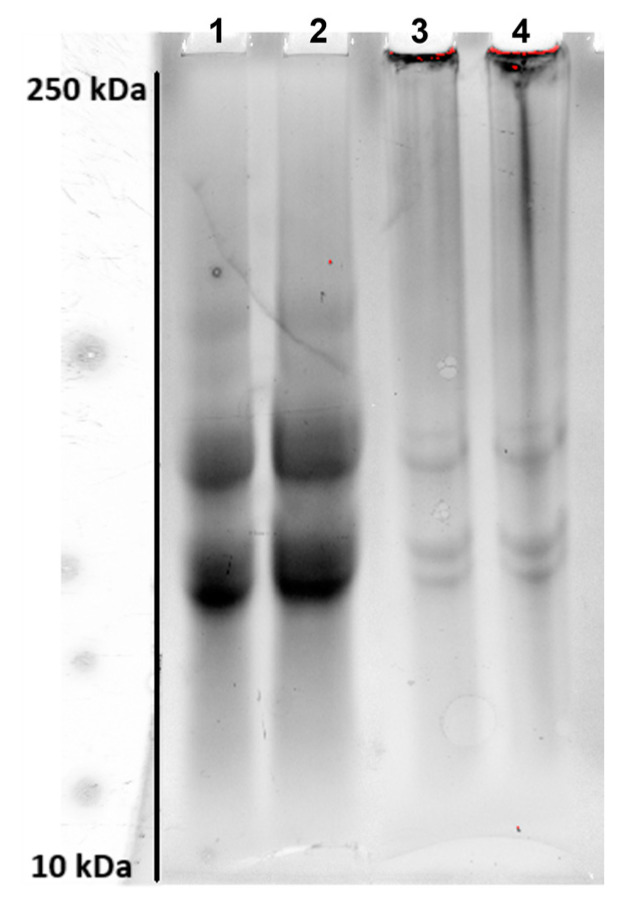
SDS-PAGE analysis of APs mixed with AM (50% (*w/w*) APs and 50% (*w/w*) AM) incubated for 2 h in the absence (1–2: 30 and 60 µg of the sample, respectively), or the presence of mTGase (40 U/g) (3–4: 30 and 60 µg of the sample, respectively).

**Figure 2 ijms-24-03405-f002:**
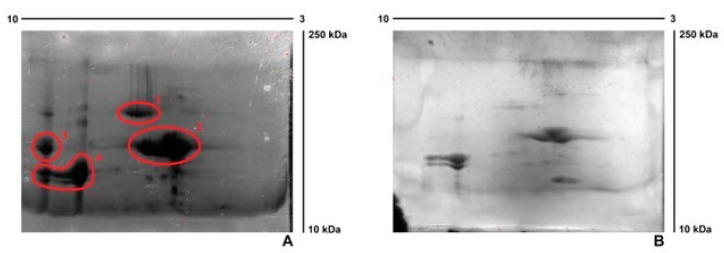
2D-PAGE of APs: samples were separated using IPG strips (3–10). (**A**) in the absence of mTGase. (**B**) in the presence of mTGase (40 U/g).

**Figure 3 ijms-24-03405-f003:**
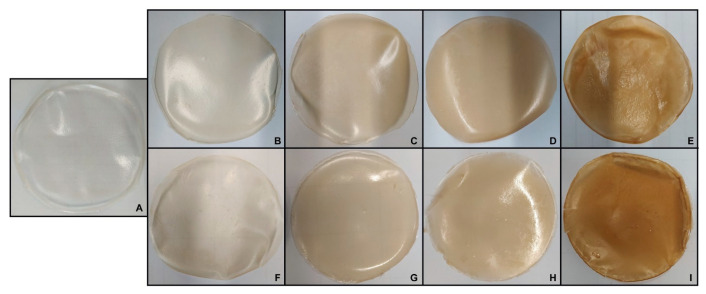
Film obtained with different ratios of APs-AM and incubated in the absence or the presence of mTGase: APs-AM(0–100) (**A**), APs-AM(15–85) in the absence of mTGase (**B**), APs-AM(15–85) in the presence of mTGase (**F**), APs-AM(30–70) in the absence of mTGase (**C**), APs-AM(30–70) in the presence of mTGase (**G**), APs-AM(50–50) in the absence of mTGase (**D**), APs-AM(50–50) in the presence of mTGase (**H**), APs-AM(100–0) in the absence of mTGase (**E**), APs-AM(100–0) in the presence of mTGase (**I**).

**Figure 4 ijms-24-03405-f004:**
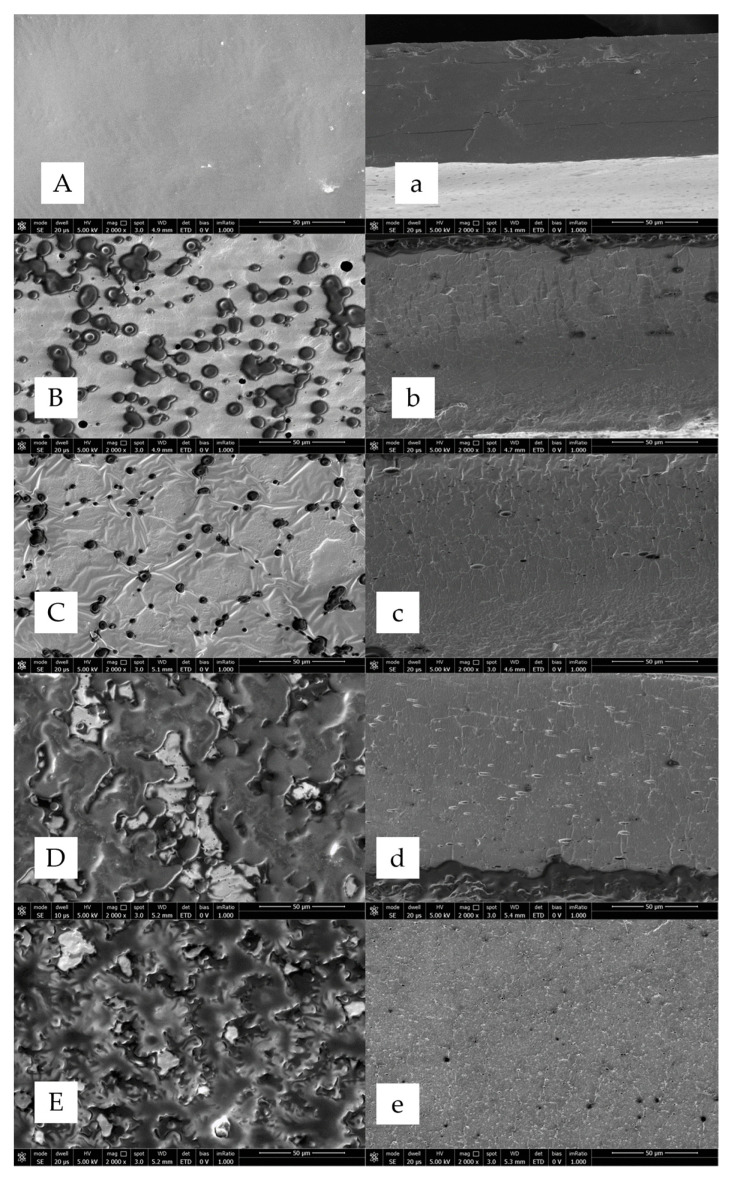
SEM micrographs (surface in uppercase and cross-section in lowercase) of AM100-based films (**A**,**a**), blended films (APs50-AM50) in the absence (**B**,**b**) and the presence of mTGase (**C**,**c**), APs100-based films in the absence (**D**,**d**) and the presence of mTGase (**E**,**e**). SEM analyses were performed at a magnification of 2000×. Further experimental details are given in the text.

**Figure 5 ijms-24-03405-f005:**
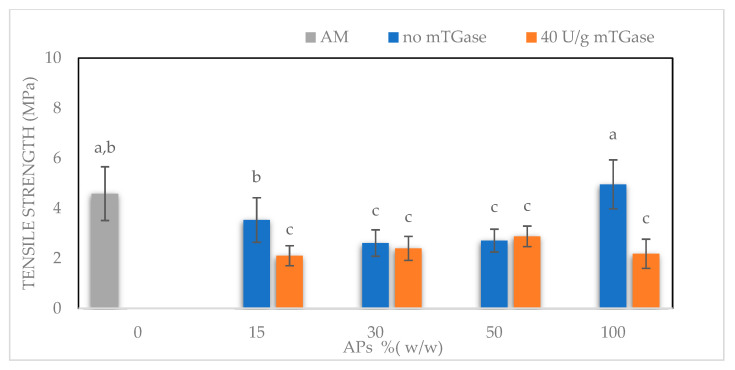
Mechanical properties of films containing 50% (*w/w*) glycerol, prepared at different ratios of APs-AM and treated or not by mTGase (40 U/g). Different small letters (a–d) indicate significant differences among the values reported in each bar (*p* < 0.05). For each sample six specimens were evaluated.

**Figure 6 ijms-24-03405-f006:**
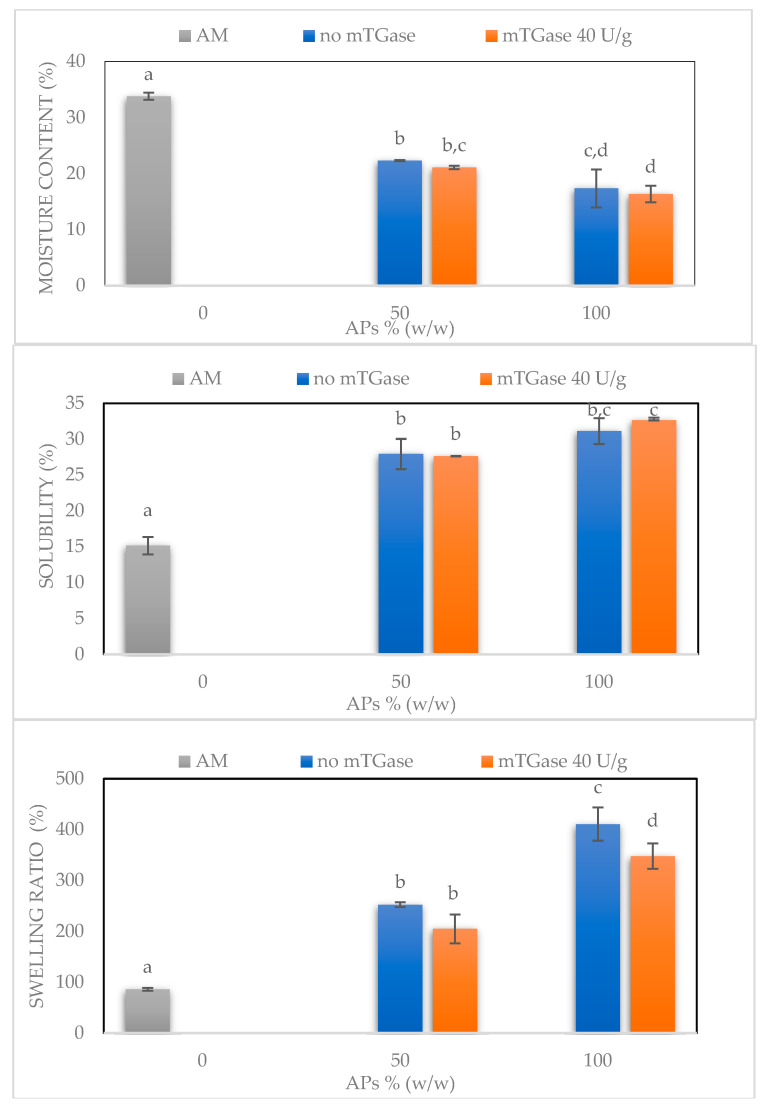
Moisture content, water solubility, and swelling ratio of AM100-based films, blended films (APs50-AM50) in the absence and the presence of mTGase, and APs100-based films treated or not by the enzyme. Different small letters (a–d) indicate significant differences among the values reported in each bar (*p* < 0.05). The analysis was carried out in triplicate.

**Figure 7 ijms-24-03405-f007:**
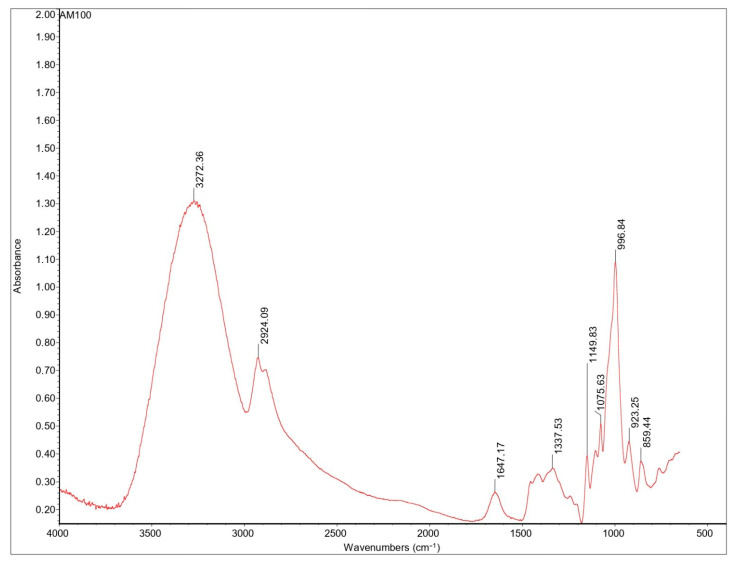
FT-IR spectra of AM100-based films.

**Figure 8 ijms-24-03405-f008:**
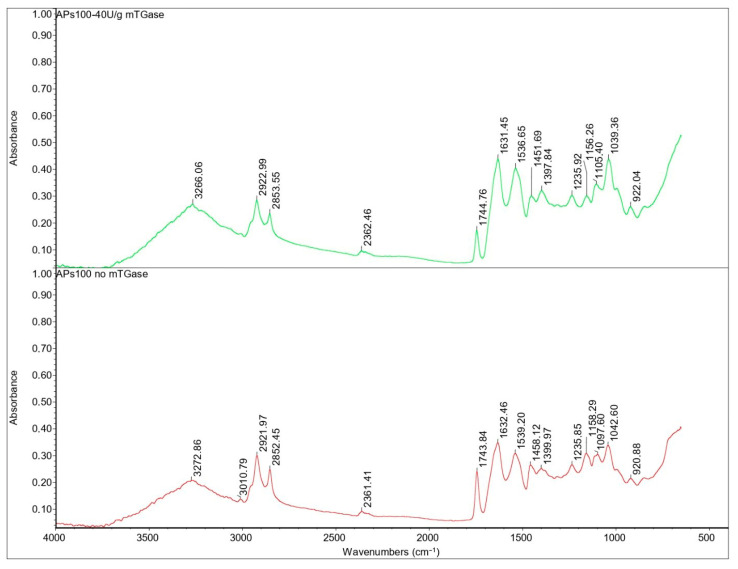
FT-IR spectra of APs100-based films in the presence and the absence of mTGase.

**Figure 9 ijms-24-03405-f009:**
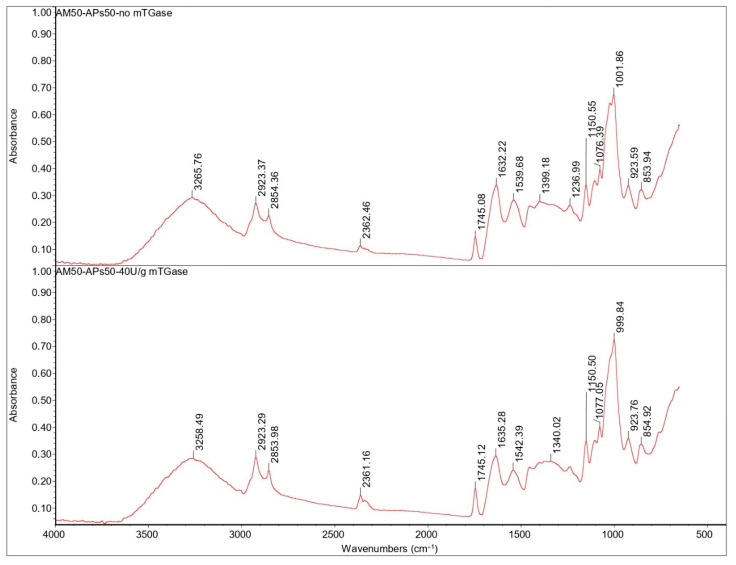
FT-IR spectra of blended films (APs50-AM50) in the absence and presence of mTGase.

**Figure 10 ijms-24-03405-f010:**
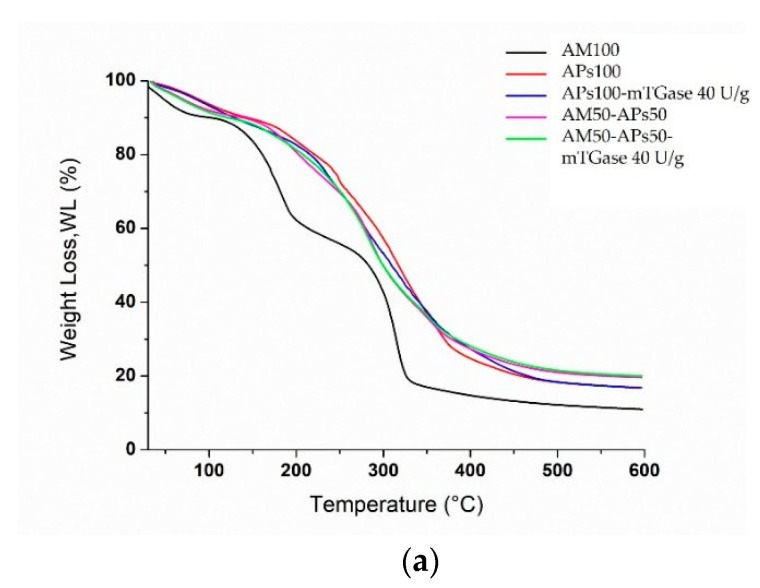
TGA (**a**) and DTG (**b**) curves of AM100-based films, blended films (APs50-AM50) treated or not by mTGase, APs100-based films in the absence and the presence of the enzyme.

**Figure 11 ijms-24-03405-f011:**
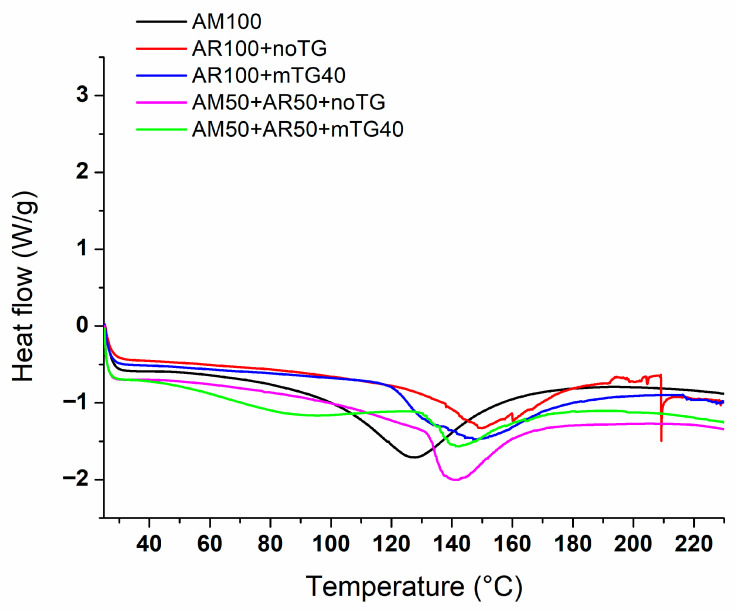
DSC curves of AM100-based films, blended films (APs50-AM50) in the absence and the presence of mTGase, APs100-based films treated or not by the enzyme.

**Table 1 ijms-24-03405-t001:** Zeta Potential analysis of FFSs at different ratios APs-AM (0–100, 15–85, 30–70, 50–50, 100–0% (*w/w*)). Different small letters (a–d) indicate significant differences among the values reported in the columns (*p* < 0.05). The analysis was carried out in triplicate.

APs-AM Ratio	Zeta Potential (mV)	APs-AM Ratio–mTGase	Zeta Potential (mV)
0–100	−10.85 ± 1.39 ^a^	-	-
15–85	−37.60 ± 0.45 ^b^	15–85	−37.58 ± 1.17 ^b^
30–70	−46.20 ± 1.20 ^d^	30–70	−39.32 ± 0.64 ^b^
50–50	−46.24 ± 1.28 ^c,d^	50–50	−42.80 ± 0.71 ^c^
100–0	−52.87 ± 1.033 ^d^	100–0	−43.23 ± 1.65 ^c^

**Table 2 ijms-24-03405-t002:** Opacity, density, and thickness of films prepared at different ratios APs-AM treated or not with mTGase (40 U/g). Different small letters (a–h) indicate significant differences among the values reported in the columns (*p* < 0.05). Analyses of opacity and density were carried out in triplicate. Thickness values are reported as the average of five values.

**APs-AM Ratio**	**Opacity** **(A_600nm_/mm)**	**Density** **(g/cm^3^)**	**Thickness** **(µm)**
0–100	1.84 ± 0.34 ^a^	1.15 ± 0.002 ^a^	102.6 ± 5.86 ^a^
15–85	3.87 ± 0.02 ^d^	1.17 ± 0.05 ^a^	111.8 ± 7.01 ^a,b^
30–70	4.45 ± 0.09 ^e^	1.18 ± 0.09 ^a^	115 ± 16.4 ^a,b^
50–50	5.28 ± 0.16 ^g^	1.20 ± 0.10 ^a^	125.6 ± 12.05 ^b^
100–0	5.55 ± 0.18 ^h^	1.25 ± 0.04 ^a^	189.4 ± 25.2 ^d^
**APs-AM Ratio—mTGase**	**Opacity** **(A_600nm_/mm)**	**Density** **(g/cm^3^)**	**Thickness** **(µm)**
15–85	2.86 ± 0.11 ^b^	1.17 ± 0.02 ^a^	114.4 ± 5.98 ^a,b^
30–70	3.04 ± 0.07 ^b^	1.20 ± 0.17 ^a^	117.5 ± 12.23 ^a,b^
50–50	3.57 ± 0.01 ^c^	1.24 ± 0.14 ^a^	149.4 ± 1.34 ^c^
100–0	4.79 ± 0.04 ^f^	1.29 ± 0.02 ^a^	225.6 ± 22.34 ^e^

**Table 3 ijms-24-03405-t003:** Water vapor permeability and CO_2_ permeability of films with different ratios of APs-AM treated or not with mTGase. Different small letters (a–d) indicate significant differences among the values reported in each column (*p <* 0.05). The analysis was carried out in duplicate.

**APs-AM Ratio**	**WVP** **(g*mm*m^−2^*d^−1^*kPa^−1^)**	**CO_2_P** **(cm^3^*mm*m^−2^*d^−1^*kPa^−1^)**
0–100	8.68 ± 0.36 ^a^	0.42 ± 0.04 ^a^
15–85	6.16 ± 0.10 ^b^	0.42 ± 0.06 ^a^
30–70	3.56 ± 0.57 ^c^	0.25 ± 0.02 ^a,b,c^
50–50	3.27 ± 0.31 ^c^	0.24 ± 0.06 ^a,b^
100–0	2.89 ± 0.13 ^c^	0.30 ± 0.09 ^a,b,c^
**APs-AM Ratio—mTGase**	**WVP** **(g*mm*m^−2^*d^−1^*kPa^−1^)**	**CO_2_P** **(cm^3^*mm*m^−2^*d^−1^*kPa^−1^)**
15–85	3.010 ± 0.03 ^c^	0.35 ± 0.02 ^a,b^
30–70	2.74 ± 0.42 ^c^	0.12 ± 0.02 ^c^
50–50	2.35 ± 0.97 ^c,d^	0.16 ± 0.04 ^c^
100–0	1.87 ± 0.28 ^d^	0.45 ± 0.07 ^a^

## Data Availability

The data presented in this study are available on request from the corresponding author.

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
