# Peer review of "Mechanical, Barrier and Thermal Properties of Amylose-Argan Proteins-Based Bioplastics in the Presence of Transglutaminase"

_ijms, 2023, doi:10.3390/ijms24043405_

Round 1
Reviewer 1 Report
This manuscript investigates the effect of adding mTGase and different ratios of Argan seed proteins and amyose in the generation and mechanical properties of thermoset biofilms.
The work described is of high interest for the development of new bioplastics and it is novel, however since the addition of TGase is the major factor to be studied here I would like to see the results on the spead of gelatinisating or/and curation with and without TGase, since I believe that would add a valueable insight for the use of this enzyme for the generation of films.
The results are well presented but in the discription of each figure and table it needs to be said that the value presented is an average of n=2 or 3 (as some tests are done in duplicate and others in triplicate) and =or- is the SD or the error (whatever it is the case).
Finally the conclusion needs to allude to the presence or not of TGase. Even if the conclusion is that the addition of TGase does not have a major significant effect in any of the characteristics. But it does not make any sense that the titles says that the authors want to study the effect of TGase in the mechanical characteristics of the film and then there is no conclusion about that.
Author Response
We thank the reviewer for your general opinion about our novel edible film that could be used also as bioplastics replacing the traditional petrol chemical-based ones.
We have not studied the speed of drying in detail, even though it seems that no differences in the rate of drying is relevant between mTGase-containing film and the ones without the enzyme. For the sake of clarity, our aim is to verify the effect of isopeptide bonds catalyzed by the enzyme on the characteristics of the films, as we have already verified in other papers that study the mTGase-mediated reticulation of the protein component on mechanical and permeability properties.
The “Conclusions” section has been changed mentioning the influence of the enzyme on some film characteristics, thus the title of the manuscript has not been changed.

Reviewer 2 Report
This paper investigated a type of bioplastic in terms of some basic properties. And it was declared to be useful for the development of different applications. In fact, from materials to applications, there are many links to be dealt with and many difficult problems to be solved. For example, in order to meet a certain application, the structure or composition of materials need to be changed. There are too many materials that may be useful. Finding a kind of material and measuring its basic properties, the innovation of this work needs to be improved. It is suggested to select a specific meaningful application, modify the structure or composition of the materials in the manuscript, and retest the properties of the modified materials. As long as the property test meets the requirements of the application, it is not necessary to test all the properties. In addition, too many abbreviations were used in the manuscript. Inappropriate abbreviations made readers misunderstand the real meaning that needed to be explained. More importantly, in this manuscript, too many ambiguous conclusions were expressed. For example, in the section of conclusion part “The protein content seems to influence Elongation at Break and Young’s Modulus, giving rise to less extensible and stiffer films, interesting for an application as bio-shoppers.” Additionally, part of the conclusion is unacceptable to readers. The conclusion part gave two conclusions. One is that “the presence of proteins influences permeability properties”. And another one is that “thermal properties are not greatly influenced by film composition”. In a word, in the manuscript, one conclusion was “A influences B”and another one was “C not greatly influences D”. It is strongly suggested that this kind of influence should be studied clearly. If the quantitative relationship is not easy to be explained, at least the most basic superficial relationship should be studied clearly.
Author Response
The authors thank the reviewer for his/her observations about the manuscript. In the literature many papers regarding bioplastics/edible films are reported as research carried out in our laboratories. Mainly they are hydrocolloid-based materials, such as polysaccharides and proteins modified or not by means of the enzyme microbial transglutaminase (mTGase), which acts as a reticulating agent if the proteins contain glutamine and lysine reactive residues, since TGase (E.C. 2.3.2.13) catalyzes the formation of iso-peptide bonds. We have tested in different papers, many kinds of polysaccharides and proteins of different origin and checked the basic performances, such as mechanical and barrier properties. From the characteristics we have observed, we have then suggested an use for application purposes and then studied, in separated publications, the effect of our materials on specific products. For example, Al-Asmar et al. have proved the effect of hydrocolloid-based coatings in reducing acrylamide formation in fried foods (potato chips, falafel and kobbas Al-Asmar et al. Coatings 2018, Al-Asmar et al. Coatings 2019, Al-Asmar et al. Foods, 2020, ) or as bioplastics from wrapping strawberries with the aim of tending the shelf life maintaining some nutritional values (Al-Asmar et al. Nanomaterials, 2020). The bioplastics proposed in this paper will be tested for their biodegradation in different kind of soils in the case their application would be as mulching sheets and/or bags for collecting organic garbage.
Following reviewer suggestions, we have rewritten the “conclusions” section and provided a list of the abbreviations after the “Abstract” section.
Please note that we have also revised the English language and asked an English native speaker colleague to read critically our manuscript.

Reviewer 3 Report
Dear Authors,There were minor errors, which I have outlined below:
The cited references are relevant, however less than 50% are recent publications (within the last 5 years). It is recommended to supplement the manuscript with the latest literature (from 2019-2023) which will increase the novelty of the work and improve the discussion. Please re-check item No. 40, as the authors, title, year of publication and journal are missing.
Line 208: “increases visibly the film opacity” better “noticeably increases the opacity of the film”
Line 211: The authors cited a paper (item 18 - Mirpoor, S. F., Giosafatto, C. V. L., Mariniello, L., D’Agostino, A., D’Agostino, M., Cammarota, M., ... & Porta, R. (2022). Argan (Argania spinosa L.) Seed Oil Cake as a Potential Source of Protein-Based Film Matrix for Pharmaco-Cosmetic Applications. International Journal of Molecular Sciences, 23(15), 8478.)
in which the opacity of "MaterBi® (61.92±3.55 A600nm/mm)[18]" was claimed to have been studied. Similar situation in line item 206: "commercial petrol-based plastic LDPE (low207 density polyethylene) (opacity value=1.44±0.04 A600nm/mm)[18]. Unfortunately, the parameters of these polymers were not studied in this paper, and the cited data were also not confirmed by citations from the literature. What follows is the duplication of unconfirmed facts. Please provide the real source of the cited results.
Author Response
A check for more recent publications has been done as requested. Please note that some references have been replaced with papers less old than 5 years. Former reference #40 has been corrected.
Line 208 the sentence suggested by the reviewer has been inserted instead of the original one.
In studying the opacity of our films, in the present paper we did measure LDPE and MaterBi opacity in the same conditions Mirpoor at al. did (Mariniello is a co-author of reference #18 that has #19 in the revised version) and thus we have decided not to insert the results obtained for LDPE and MaterBi opacity which were, of course, the same.
